# Co-Expression Network Analysis and Hub Gene Selection for High-Quality Fiber in Upland Cotton (*Gossypium hirsutum*) Using RNA Sequencing Analysis

**DOI:** 10.3390/genes10020119

**Published:** 2019-02-06

**Authors:** Xianyan Zou, Aiying Liu, Zhen Zhang, Qun Ge, Senmiao Fan, Wankui Gong, Junwen Li, Juwu Gong, Yuzhen Shi, Baoming Tian, Yanling Wang, Ruixian Liu, Kang Lei, Qi Zhang, Xiao Jiang, Yulong Feng, Shuya Zhang, Tingting Jia, Lipeng Zhang, Youlu Yuan, Haihong Shang

**Affiliations:** 1State Key Laboratory of Cotton Biology, Key Laboratory of Biological and Genetic Breeding of Cotton, The Ministry of Agriculture, Institute of Cotton Research, Chinese Academy of Agricultural Sciences, Anyang 455000, Henan, China; zxy1306446323@google.com (X.Z.); liuay@cricaas.com.cn (A.L.); 18003725804@126.com (Z.Z.); gejiaoyu@163.com (Q.G.); fansenmiao@caas.cn (S.F.); wkgong@aliyun.com (W.G.); junwenlee@163.com (J.L.); juwugong@126.com (J.G.); syzmb@aliyun.com (Y.S.); wangyanlin@163.com (Y.W.); ruixianliu6@126.com (R.L.); leikang2016@163.com (K.L.); qizhangyx@126.com (Q.Z.); genesisjiang@163.com (X.J.); 15939623814@163.com (Y.F.); zhangshuya1996@126.com (S.Z.); 18355452765@163.com (T.J.); z2277519159@163.com (L.Z.); 2School of Agricultural Science, Zhengzhou University, Zhengzhou 450001, Henan, China; 13247299145@163.com

**Keywords:** *Gossypium hirsutum*, fiber development, transcriptomic analysis, DEGs, WGCNA

## Abstract

Upland cotton (*Gossypium hirsutum*) is grown for its elite fiber. Understanding differential gene expression patterns during fiber development will help to identify genes associated with fiber quality. In this study, we used two recombinant inbred lines (RILs) differing in fiber quality derived from an intra-*hirsutum* population to explore expression profiling differences and identify genes associated with high-quality fiber or specific fiber-development stages using RNA sequencing. Overall, 72/27, 1137/1584, 437/393, 1019/184, and 2555/1479 differentially expressed genes were up-/down-regulated in an elite fiber line (L1) relative to a poor-quality fiber line (L2) at 10, 15, 20, 25, and 30 days post-anthesis, respectively. Three-hundred sixty-three differentially expressed genes (DEGs) between two lines were colocalized in fiber strength (FS) quantitative trait loci (QTL). Short Time-series Expression Miner (STEM) analysis discriminated seven expression profiles; gene ontology (GO) and Kyoto Encyclopedia of Genes and Genomes (KEGG) annotation were performed to identify difference in function between genes unique to L1 and L2. Co-expression network analysis detected five modules highly associated with specific fiber-development stages, especially for high-quality fiber tissues. The hub genes in each module were identified by weighted gene co-expression network analysis. Hub genes encoding actin 1, Rho GTPase-activating protein with PAK-box, TPX2 protein, bHLH transcription factor, and leucine-rich repeat receptor-like protein kinase were identified. Correlation networks revealed considerable interaction among the hub genes, transcription factors, and other genes.

## 1. Introduction

Cotton (*Gossypium* spp.) is among the most important crops worldwide and is grown for its natural fiber for the textile industry. The global area planted with *Gossypium hirsutum* (upland cotton) comprises 95% of all planted cotton because of its long, strong, and white fibers and high yield [1]. With the dramatic changes in climate and environment and the reason of the often crosspollinated plant, breeders are focused on the development of new cultivars to improve the fiber quality of upland cotton, which will be highly beneficial for the textile industry.

A thorough understanding of the underlying mechanisms of cotton fiber development is crucial to improve fiber quality. Fiber development can be classified into five stages: Fiber initiation (0–3 days post-anthesis (DPA)), elongation (3–15 DPA), transitional cell-wall remodeling (15–20 DPA), secondary wall biosynthesis (20–40 DPA), and maturity (40–50 DPA) [2]. The first two stages determine the number and length of the fibers. The third and fourth stages are associated with fiber strength and fineness as a result of the development of secondary cell wall thickening.

Comparative transcriptome sequencing is an efficient technology useful for differential expression analysis to identify candidate genes associated with target traits. Cotton fibers can be sampled from cotton bolls at each development stage. RNA sequencing (RNA-seq) provides a means of analyzing individual gene transcription and the entire transcriptome profile throughout fiber development. In the past decade, numerous comparative transcriptome analyses of upland cotton utilized chromosome segment substitution lines derived from *G. hirsutum* × *Gossypium barbadense* [3,4,5], wild and domesticated *G. hirsutum* [1,6], and near-isogenic lines [7,8], which discovered many candidate genes associated with high-quality fiber. However, there are few studies of gene expression patterns using a recombinant inbred line (RIL) population in which the progeny lines are approximately homozygous. In our previous study, 16 stable quantitative trait loci (QTLs) for fiber strength were identified from a RIL population consisting of 196 lines [9], in which certain individuals showed exceptionally high-quality fiber strength. To explore and identify genes responsible for the difference in fiber quality among the progeny and to compare gene expression profiles between high- and low-quality fiber lines, RNA-seq analysis was performed in the present study using two RILs that differed in fiber quality.

## 2. Methods

### 2.1. Plant Materials

Two upland cotton lines designated 69307 (L1) and 69362 (L2) from an intraspecific F_6:8_ RIL population were used. The RIL population was developed from a cross between the strains ‘0-153’ and ‘sGK9708’, which are characterized by excellent fiber quality and high yield, respectively. The procedure for development of the RIL population was detailed previously [9,10,11]. The high-quality fiber line L1 is a positive extreme-parent line, whereas L2 is a negative extreme-parent line for fiber quality [12].

The two lines were used as plant materials and grown under standard field conditions at the Experimental Station, Institute of Cotton Research of the Chinese Academy of Agricultural Sciences located in Anyang, Henan, China in 2015. The first day of flowering was recorded as 0 DPA and flower buds were tagged. At 10, 15, 20, 25, and 30 DPA cotton fibers were collected from bolls, with three biological repeats, using a sterile medical scalpel and were frozen in liquid nitrogen for RNA-seq analysis. The corresponding samples for the L1 and L2 lines were designated L1D10, L1D15, L1D20, L1D25, L1D30, L2D10, L2D15, L2D20, L2D25, and L2D30, where D is days post-anthesis. RNA-Seq raw data accession number SRP172799 in NCBI sequence read archive (http://www.ncbi.nlm.nih.gov/sra/) and bioproject archive (http://www.ncbi.nlm.nih.gov/bioproject/) PRJNA508480.

### 2.2. RNA Isolation, Library Construction, and RNA-Seq Analysis

Total RNA was extracted from each fiber sample using the RNAprep Pure Plant Kit (Polysaccharides & Polyphenolics-rich, Tiangen, Beijing, China) following the manufacturer’s protocol. The RNA was quantified using a NanoDrop 2000 spectrophotometer (Thermo Scientific, Waltham, MA, USA). RNA degradation and contamination were assessed using 1% agarose gel electrophoresis. The RNA integrity was confirmed with an Agilent 2100 Bioanalyzer (Agilent Technologies, Santa Clara, California, USA). A total amount of 2 μg RNA for each sample was used for transcriptome library sequencing. Libraries were constructed using the Illumina TruSeq™ RNA Sample Preparation Kit (Illumina, San Diego, CA, USA) following the manufacturer’s recommendations. A total of 30 libraries for the two cotton lines at five developmental stages (10, 15, 20, 25, and 30 DPA) with three biological repeats were sequenced using an Illumina HiSeq 2500 sequencing platform with 125 base pair (bp) paired-end (PE) raw reads (Berry Genomics Co., Ltd., Beijing, China).

Raw data in Fastq format were processed using Trimmomatic software [13]. Clean data were obtained by removing reads that contained the adapter, poly-N, and low-quality reads, which were reads with ≥10% unidentified nucleotides (N), >50% bases with Phred quality < 5, >10 nt aligned to the adapter, and 0 nt aligned to the adapter. Finally, the percentage GC content and Q30 were determined for the clean data, which were then used for downstream analysis. Bowtie v2.0.6 [14] was used to build an index of the reference genome, which was downloaded from the CottonGen database (http://www.cottongen.org) [15]. The clean data were mapped to the *G. hirsutum* genomes [16,17] with default parameters using TopHat2 [18]. The fragments per kilobase of exon per million reads (FPKM) values of genes were calculated using Cufflinks [19]. Pearson correlation analysis was performed to calculate correlation coefficients between samples. Samples with correlation coefficients less than 0.8 among the three biological replicates were removed from the data set.

### 2.3. Differentially Expressed Genes Analysis and Comparison of Differentially Expressed Genes Expression Patterns

Differentially expressed genes (DEGs) were identified using the DESeq2 R package using the count number of each gene [20,21]. The DEGs were identified with an expression level FPKM > 0.5, false discovery rate < 0.01, and log_2_ fold change value > 1 or <−1 between each pairwise comparison.

Given that different gene expression patterns might result in different phenotypes, the Short Time-series Expression Miner (STEM) software was used to explore the expression patterns of DEGs during fiber development [22,23]. To understand function of the DEGs, the gene ontology (GO) enrichment was performed using BLASTX program [24] and GO databases (http://archive.geneontology.org/latest-lite/) and (ftp://ftp.ncbi.nlm.nih.gov/gene/DATA/); and Kyoto Encyclopedia of Genes and Genomes (KEGG) enrichment analysis was carried out using KOBAS 3.0 software [25,26].

### 2.4. Construction of Gene Co-Expression Networks and Screening of Hub Genes 

The WGCNA (weighted gene co-expression network analysis) R package [27] was used to analyze co-expression of genes and select highly correlated (or hub) genes that may be strongly associated with fiber quality. The genes clustered in modules that contained a coefficient (>0.6) with each sample, especially the elite fiber tissues, were used to further analysis. After completion of the co-expression analysis, the edge files were sorted by weight value and the first 200 pairs of network connections were used to establish interaction networks among the DEGs. The hub genes were screened on the basis of module membership (*K*_ME_) values. The interaction networks among were drawn using Cytoscape 3.6.1 [28].

## 3. Results

### 3.1. Transcriptome Sequencing Analysis and Correlation of Replicate Samples

To identify genes that play crucial roles in fiber development, fibers were sampled at five developmental stages (10, 15, 20, 25, and 30 DPA) and transcriptome sequencing was conducted. A total of 1543.602 million clean reads were obtained from 30 libraries, with an average of 51.45 million reads per sample. In the reads mapping analysis, 91.56% to 95.98% reads could be mapped to the reference genome. The Q30 was 89.45% to 94.65%, with a mean value of 93.45%, and the GC content range was 44.22% to 47.98%, which indicated that the RNA-seq data were reliable (Appendix A). To evaluate the distribution density of clean data reads, reads were mapped to the 26 chromosomes of *G. hirsutum* using Bowtie 1 and visualized using the Circos program [29] (Appendix A), which indicated the difference of levels of transcripts among the fiber development stages.

After reads mapping and FPKM calling, the correlation between biological replicates of samples was assessed (Appendix A). In this analysis, genes with FPKM > 0.5 were considered to be expressed genes. Consequently, 41,583, 40,429, 40,236, 39,544, and 39,032 genes were expressed at 10, 15, 20, 25, and 30 DPA in the L1 fiber tissues, respectively. Similarly, 41,711, 40,463, 40,532, 40,929, and 39,561 expressed genes were detected at the five respective stages in L2 fiber tissues. Genes with FPKM values of 0.5–5, 5–100, and >100 accounted for 45.4%, 49.7%, and 4.9% of all gene models, respectively (Figure 1). 

### 3.2. Differential Gene Expression Analysis

To explore the genes associated with fiber quality in fiber tissues, the DEGs were identified in vertical and horizontal comparisons, i.e. between the same developmental stage of the two lines and between different developmental stages of each line. The numbers of DEGs detected are shown in Figure 2. After removal of duplicate genes, a total of 13,881 DEGs were identified and their function descriptions annotated in vertical (Appendix A) and horizontal (Appendix A) comparisons, respectively. The FPKM values of the 13,881 DEGs in each sample are shown in Appendix A.

At 10 DPA, 99 genes were significantly differentially expressed, including 72 upregulated and 27 downregulated genes. Representative DEGs that not only showed high |log_2_(FC)| between L1 and L2, but also significantly (corrected *p*-value < 0.01)) enriched in the KEGG pathways are listed in Table 1 and Table 2, respectively. In upregulated genes, two brassinosteroid biosynthesis pathway genes encoding cytochrome P450 proteins exhibited a five-fold change in expression in the high-quality fiber line L1 compared with that of L2, which may enhance plant performance and productivity [30]. A gene of the auxin-responsive GH3 protein family, which participates in auxin homeostasis by catalyzing auxin conjugation and binding free indole-3-acetic acid to amino acids [31], showed an approximately three-fold increase in expression in L1. Additional genes, such as genes encoding a glycosylic hydrolase protein, a 10-formyltetrahydrofolate synthetase, and a UDP-glucose 6-dehydrogenase protein, also showed several-fold higher expression levels in L1 compared with those in L2. In the downregulated genes, only six genes were enriched into four pathways listed in Table 2. Interestingly, the expression levels of two transcription factors encoding WRKY11, which has been characterized to be involved in the response to dehydration stress [32] and plays key role in plant defense [33], and C3H proteins in L1D10 were four times than that of L2D10.

At 15 DPA, 1137 and 1584 genes were significantly upregulated and downregulated, respectively, in fiber tissue of the high-quality L1 line compared with their expression levels in the L2 line. A total of 830 DEGs were detected between the L1D20 and L2D20 fiber samples, of which 437 genes were upregulated and 393 genes were downregulated in the L1. Representative upregulated and downregulated genes are listed in Table 3 and Table 4, respectively. The expression levels of two acyl-CoA oxidase genes and a beta-ketoacyl reductase 1 gene involved in the biosynthesis pathway for unsaturated fatty acids exhibited three-fold changes in expression in the L1D20 sample compared with their expression levels in the L2D20 sample. Seven genes associated with microtubule development in the phagosome pathway showed 2.4- to 8.9-fold changes in expression level in L1 compared with those of L2 at 20 DPA. Genes encoding ketoacyl-CoA synthase, 3-ketoacyl-acyl carrier protein synthase I, and acetyl-CoA carboxylase carboxyl transferase subunit beta in the fatty acid elongation pathway, which are essential for membrane biosynthesis [34], were also enriched. In addition, transcription factors, such as basic helix-loop-helix (bHLH), bZIP and CCCH-type zinc finger protein, MYB-like Ap2/B3 protein, and WRKY protein, showed their highest expression level in the L1D20 fiber sample. In the downregulated genes, the expression of four genes encoding cytokinin oxidase in L2D20 exhibited 3.2~4.6-fold changes compared with those of L1D20, in which the high expression of cytokinin oxidase may lead to the degradation of cytokinin, thus reducing seed yield [35]. In addition, transcription factors, such as GRAS that might promote the stress-resistant [36] bZIP, myb-like .etc were also expressed higher in L2D20 than that of L1D20.

After the transitional cell-wall remodeling stage, 1019 and 184 genes were significantly upregulated and downregulated at 25 DPA, respectively. At the fiber maturity stage, 4034 genes were significantly differentially expressed in pairwise comparisons between L1 and L2 at 30 DPA, of which 2555 and 1479 genes were upregulated and downregulated at 30 DPA, respectively.

### 3.3. Congruence Analysis with the Previous Report

In a previous study, Zhang et al. [9] identified 16 stable QTLs for fiber strength using the RILs of present study, in which 3364 genes were identified as candidate genes by comparison with the reference *G. hirsutum* genome. To identify genes that were not only associated with fiber strength in the entire RILs population, but also were differentially expressed in progeny of the high-quality and poor-quality fiber lines, we compared these candidate genes and the DEGs between L1 and L2. Of the 3364 genes in total, 363 DEGs comprising 4, 75, 39, 62, and 183 genes at 10, 15, 20, 25, and 30 DPA were detected, of which 228 (62.6%) genes were upregulated in L1. These genes that were not only located in QTLs related to FS trait, but also identified to be differentially expressed in the two lines might contribute to the phenotypic differences in FS. In these genes, the gene *Gh_A07G1758* encoding RAB GTPase homolog B1C is required for starch and sugar homeostasis in *A. thaliana.* Meanwhile, *Gh_A07G1758* was also detected to be associated with fiber FL and FS by GWAS method [37,38]. The genes *Gh_D06G1939* encoding WRKY DNA-binding protein could enhance the drought and salt stress tolerance of transgenic *Nicotiana benthamiana* [38]. Another gene, *Gh_D06G1944*, encodes a putative leucine-rich repeat receptor protein kinase, which is characterized to control pollen production of the *Gossypium* anther [39]. The functional annotation of these genes in *Arabidopsis* is shown in Appendix A.

### 3.4. Temporal Gene Expression Patterns

To explore temporal differences in gene expression profiles in the two lines, STEM analysis of the DEGs was performed. A total of 8506 and 8415 genes were identified and organized into seven expression profiles (Figure 3a) with e-values less than 0.001 in L1 and L2. Each profile contained a set of genes that showed similar expression patterns at all fiber development stages (Appendix A). The largest two gene sets were profile ID 4 and 22, which represented a continuous upregulated and downregulated trend, respectively, as fiber development progressed. The expression profile 22 contained 2308 genes in L1, which was 19.7% more than the number of genes for L2 (1928 genes). Comparison of the genes of the two profiles between L1 and L2 was visualized by means of a Venn diagram (Figure 3b,c), which showed that approximately half of the genes were different in the expression pattern 4, even though the number of genes was similar. A total of 1112 and 1060 genes were specific to L1 and L2, respectively, for expression profile 4, with 1298 genes in common. Similarly, 1180 genes were common to L1 and L2 for expression profile 22, with 1128 and 748 genes respectively unique to each line.

These differences in gene expression may contribute to phenotypic variation. Hence, we performed GO enrichment analysis of the genes in profiles 4 and 22 for each of L1 and L2. The 30 most frequent GO terms are shown in Figure 4 and Figure 5. The terms were categorized into biological process, cellular component, and molecular function, which showed differences between the two RIL lines. In L1_profile 4, genes were annotated with the GO terms “regulation of transcription, DNA-templated” (112 genes, 10.1%), “transcription, DNA-templated” (94 genes, 8.5%), “protein phosphorylation” (92 genes, 8.3%), and “response to abscisic acid” (91 genes, 8.2%), which differed from that of L2_profile 4. In the molecular function category, the GO terms “transcription factor activity, sequence-specific DNA binding” (98 genes in L1, 8.8%) and “protein serine/threonine kinase activity” (91 genes in L1, 8.2%) showed difference between the two lines (Figure 4). The GO terms for the L1_profile 22 were mainly of the cellular component category, differing from the L2_profile 22, which was manifested as “endosome” (113 genes, 10%), “trans-Golgi network” (110 genes, 9.8%), “Golgi membrane” (98 genes, 8.7%), and “vacuolar membrane” (77 genes, 6.8%) (Figure 5).

To investigate differences in the gene pathways involved in the profiles 22 and 4, KEGG enrichment analysis was performed under the background of *Arabidopsis thaliana* and pathways are shown in a bubble graph (Figure 3d,e). The genes in expression profile 22 were mainly enriched in metabolic pathways and biosynthesis of secondary metabolites, followed by plant hormone signal transduction, phenylpropanoid biosynthesis, and cysteine and methionine metabolism. In contrast, in expression profile 4, in addition to enrichment in metabolic pathways and biosynthesis of secondary metabolites, genes were enriched in ribosome, phagosome, oxidative phosphorylation, and carbon metabolism pathways.

Other expression profiles, such as profiles 23 and 10, showed specific expression patterns at 20 DPA and 25–30 DPA, which may indicate that the expressed genes were associated with cell-wall remodeling and secondary cell-wall synthesis. The enriched KEGG pathways for these profiles are shown in Appendix A. The genes enriched in profile 23 were annotated with the terms “amino sugar and nucleotide sugar metabolism”, including 10 genes that encode galacturonosyltransferases, and “starch and sucrose metabolism” consisting of sucrose synthases (*Gh_D07G0139* and *Gh_D06G0832*), which are expressed in response to cotton fiber growth and development [40], and UDP-D-glucuronate 4-epimerase (*Gh_D07G1467*), which plays a role in providing UDP-D-galacturonate to glycosyltransferases in pectic polysaccharide synthesis [41]. The enriched pathways “plant hormone signal transduction”, including a GRAS family transcription factor that is valuable for stress-resistance breeding in *G. hirsutum* [36], “cutin, suberin, and wax biosynthesis” (7 genes), “metabolic pathways” (121 genes) were identified by genes of expression profile 10, which indicated that these regulatory pathways played key roles in secondary cell wall growth of fiber tissues (Appendix A).

In addition, the functional differences of genes unique to L1 and L2 in the same expression trend of profile 10 and 23, respectively, were also performed using the GO enrichment analysis. In profile 10, there are 148 common genes with 628 and 571 genes were unique to L1 and L2, respectively (Appendix A). The GO terms of the unique genes were shown in Appendix A, which showed the differences in terms of “oxidation-reduction process” (40 genes, 6.37%), “response to water deprivation” (38 genes, 6%), “defense response to bacterium” (32 genes, 5.1%) in biological process category of L1. Three terms in cellular component such as “Golgi apparatus” (45 genes, 7.17%), “endoplasmic reticulum” (38 genes, 6.05%) and “chloroplast stroma” (31 genes, 4.94%) in L1 were enriched. Totally 189 common genes showed the same expression profile 23 with 723 and 689 genes unique to L1 and L2, respectively (Appendix A). The top 30 enrichment terms of these genes were showed in Figure 4b,c. For example, terms “response to salt stress” (60 genes, 8.3%), “transcription, DNA-templated” (48 genes, 4.64%) and “oxidation-reduction process” (43 genes, 5.95%) in category of biological process were different between L1 and L2. 

### 3.5. Gene Co-Expression Network Analysis

To gain further insight into the relationship between gene expression and progressive fiber development and to identify genes that were specifically associated with fiber quality, the co-expression networks for a total of 13,881 differentially expressed genes (DEGs) was generated using weighted gene co-expression network analysis (WGCNA). 

The topology overlap matrix was built using the hierarchical clustering method and the dynamic cut module that characterized similar expression patterns were merged. In total, 13 gene modules were identified based on genes expression profiles and lines, of which five modules (indicated in red in Figure 6) were highly associated with an individual developmental stage of both L1 and L2 or were specifically associated with certain developmental stage of the high-quality fiber line on the basis of their coefficient between modules and tissues (Figure 6, Appendix A). 

During the fiber elongation stage, the dark red module contained 1370 genes, including 84 transcription factors associated with the sample at 10 DPA in both L1 and L2. A total of 657 genes (48 transcription factors) were clustered in the dark turquoise module, which was specific to the high-quality fiber L1 line at 15 DPA. The violet module (178 genes) was significantly specific to the L1 phenotype at 20 DPA, which was the period of transitional cell-wall remodeling or synthesis of the winding layer [42]. The pink module harbored 443 genes that were not only highly associated with L1 at 30 DPA, but also associated with L1 at 20 DPA, which indicated that these genes were involved in both transitional cell-wall remodeling and fiber maturity. The dark grey module contained 1593 genes and was also significantly associated with L1 at 30 DPA. Eighty-four of the 1593 genes were transcription factors, which suggested that these genes played key roles in the shaping of fiber quality. 

A KEGG pathway analysis of the genes in each module was performed. The expression patterns of representative genes in key pathways and the FPKM values are shown in Figure 7 and Appendix A. Genes in pathways for ubiquinone and other terpenoid-quinone biosynthesis, fatty acid metabolism, and phenylpropanoid biosynthesis were highly expressed at 10 DPA in L1 and L2. Four genes, namely *Gh_D05G0184*, *Gh_A05G0122*, *Gh_A02G1611*, and *Gh_D06G1792*, were annotated with the brassinosteroid biosynthesis pathway, which is required for fiber initiation as well as fiber elongation [43]. The genes in the dark turquoise module were mainly associated with photosynthesis, photosynthesis-antenna proteins, and galactose metabolism, which indicated their involvement in the accumulation of organic materials. In addition, genes in the dark grey module were associated with biosynthesis of secondary metabolites, cutin, suberin and wax biosynthesis, and fatty acid metabolism.

### 3.6. Identification of Hub Genes for Elite Fiber Quality and Visualization of Correlation Networks

In WGCNA, *K*_ME_ values indicate the eigengene connectivity, based on the assumption that hub genes are identified by sorting *K*_ME_ values. In the present study, the several top-ranked genes with the highest *K*_ME_ values in each of the five modules were selected as hub genes and are listed in Table 5. The hub genes in the dark red module encode actin proteins, glycosyl hydrolase proteins, and Rho GTPase-activating proteins with a PAK-box, which may significantly affect fiber elongation. Hub genes highly associated with L1 at 15 DPA were annotated as targeting protein for the Xklp2 (TPX2) protein family, and calcium-dependent phosphotriesterase superfamily protein.

Given that 10 DPA and 20 DPA were the critical fiber developmental stages of primary cell wall (PCW) and the transition to secondary cell wall (SCW) growth [1,2,44], respectively, the correlation networks of the high-weight values pairs were visualized using Cytoscape software. The candidate hub genes in the dark red module at 10 DPA were highlighted in red in the networks (Figure 8a); similarly, the hub genes in the violet module that were highly associated with 20 DPA of the high-quality fiber line (L1) are shown in Figure 8b. The gene *Gh_D10G0295* is both a hub gene and a bHLH DNA-binding protein 30 [45].

## 4. Discussion

### 4.1. Transcriptome Sequencing of Elite and Poor-Quality Fiber Lines Mapped to the Reference Genome of Upland Cotton

RNA-seq could be a useful tool to discover whole-genome expression profile and DEGs for screening candidate genes related to fiber development. In the past decade, application of RNA-seq were reported in fiber development [1,3,4,7,57], in biotic stress [58,59,60], and in abiotic stress [61,62]. In the present study, five time points (10, 15, 20, 25, and 30 DPA) were chosen to investigate the genetic difference in fiber development between the progenies of two extreme-parent RILs. The 10 and 15 DPA time points coincided with critical stage for fiber elongation, which may continue up to 25 DPA [2,63]. Cotton fiber tissues start transitional cell wall remodeling at 20 DPA, which represents the onset of the secondary wall synthesis [42]. The 25 DPA and 30 DPA time points represented the period of secondary wall thickening, which affects fiber strength and fineness [44]. Therefore, the DEGs identified between the two lines during fiber development, and mapped to the published reference genome and annotated transcripts for *G. hirsutum* [16,17], may reflect the genetic differences associated with fiber quality. The average number of clean reads and Q30 were 51.45 million and 93.8% per sample, respectively, which were indicators of the reliable quality of the RNA-seq data. However, five samples in the third biological replicate showed low correlation (<0.8), which may be a result of weather conditions or another unknown reason.

### 4.2. RNA-Seq Provides Potentia information by Comparison of Expression Profiles between the Two Recombinant Inbred Lines 

Vertical comparison of DEGs showed that a large number of genes were significantly expressed in L1 and L2 at each stage of fiber development, which revealed that gene expression patterns differed considerably between the two lines. DEGs analysis of lines showing different phenotypes can uncover functional genes related to target traits. Previous studies for exploring fiber quality and yield and resistance to *Verticillium wilt* were also performed using RNA-seq. Islam et al. performed RNA-seq of fiber tissues on 15 and 20 DPA, which showed that two signaling pathways including ethylene and the interconnected phytohormonal pathways and receptor-like kinases (RLKs) signaling pathways might contribute to the high fiber strength [7]. RNA-seq focusing on fiber length was also performed using the backcross inbred lines on 5, 10, 15, and 20 DPA, which identified that eight DEGs were located in the FL QTLs and that three SNP markers were associated with FL trait [3]. A SKIP35 protein involved in ubiquitin-mediated signal was identified using RNA-seq and validated to enhance the tolerance to *Verticillium wilt* [58]. 

In the present study, upregulated and downregulated genes between two lines were identified in each stage, and representative DEGs were partly characterized in *A. thaliana* and *Gossypium*. The functions of these genes that played key roles during plant development in *A. thaliana* still needed to be solved in *Gossypium.* For instances, an auxin-responsive GH3 family protein enriched in plant hormone signal transduction pathway participates in auxin homeostasis [31]. A WRKY transcription factor plays roles in dehydration stress and plant defense in upland cotton [32]. Interestingly, at 10 DPA, the genes *Gh_D05G0184* and *Gh_A05G0122* were significantly upregulated in L1. These genes are involved in the brassinosteroid pathway and encode cytochrome P450 proteins, playing roles in catabolizing active BRs during fiber elongation by modulating ethylene and cadmium signaling [64]. The tubulin proteins enriched in the “phagosome” pathway were highly upregulated in L1 compared with the expression level in L2 at 20 DPA (Table 3), and are involved in cotton fiber development [65]. In each stage, part of DEGs were colocalized in stable FS QTLs, meanwhile, several genes listed in Section 3.3 could be identified in previous study [37,38], which suggests that these DEGs are more valuable for influencing fiber strength. 

The similar expression trends but different gene sets (Appendix A) and further GO analysis of genes unique to L1 and L2 in expression profile 4, 22, 10, and 23, as shown in Figure 4 and Figure 5, revealed the difference between the two lines, which contribute to the difference in FS. It is noteworthy that genes clustered in expression profile 23 and 10 were specific expressed in secondary cell wall thickening and cell wall remodeling. For instance, genes participate in “amino sugar and nucleotide sugar metabolism” pathway can affect the cotton fiber growth and development [40], as detailed in Section 3.4. The specific function of these genes in *Gossypium* remains to be studied.

### 4.3. Hub Genes Identified by WGCNA May Play Critical Roles in Improvement of Fiber Quality

In the present study, we used WGCNA to identify significant modules of genes associated with specific developmental stages, especially for the high-quality fiber line L1, and to select hub genes. Five modules were highly associated with high-quality fiber, hence further analysis of these modules was performed in the present study, although several modules also showed high correlation with samples of the poor-quality fiber line L2. Transcription factors play key roles in plant growth and development. For instance, the transcription factors in elongation stage at 10 DPA contained 11 HD-ZIP that play an important role in epidermal development [66], nine bHLH that are involved in brassinosteroid (BR) signaling in fiber development of upland cotton [67], six ERF responded to salt/drought and ABA [68], and each five of MYB, C2H2, C3H, and ARF (Appendix A). On 30 DPA, the transcription factors of TALE, MYB, GRAS, NAC, and Trihelix GATA were specific expressed in fiber cell development. Several KEGG pathways were highly specific to fiber developmental stages such as “ubiquinone and other terpenoid-quinone biosynthesis” at 10 DPA, “brasssinosteroid biosynthesis” and “plant hormone signal transduction at 15 DPA of L1, photosynthesis at 20 DPA of L1 .etc played are important factors in affecting fiber development. The functions of hub genes were partly characterized in *A. thaliana* (Table 5). A TPX2 protein was identified as hub gene. In *A. thaliana*, TPX2 protein performs multiple roles in microtubule organization such as reinforcing microtubule formation in the vicinity of chromatin and the nuclear envelope [69]. A WDL3 protein belonging to TPX2 family was in response to light to modulate hypocotyl cell elongation [47]. The interactions between hub genes and transcription factors could be a kind of transcriptional regulation. An actin 1 gene (*Gh_A10G1961*) was predicted to interact with five transcription factors (2 SBP, NAC, GRF and GATA). Therefore, these hub genes were also worthy of further research.

## 5. Conclusions

In summary, to understand the differences in gene expression patterns between two extreme-parent lines derived from an intra-*hirsutum* RIL population of upland cotton and to identify genes associated with elite fiber quality, RNA-seq was performed by constructing 30 libraries from the two lines with three biological replications at five time points that coincided with the stages of fiber initiation, elongation, cell-wall remodeling, and secondary cell wall growth. Hence, the DEGs were identified by multiple comparisons, especially the vertical comparison between L1 and L2, and the representative genes showed key functions in fiber elongation and thickening. The 363 genes identified by RNA-seq and colocalized in FS QTL were valuable for affecting fiber development. Gene ontology analysis underlined the difference in gene function categories between two key expression patterns (profiles 4 and 22; continuous upregulation and downregulation, respectively), and another two expression profiles (profiles 10 and 23) that were highly associated with fiber cell wall remodeling and thickening. KEGG analysis revealed that the genes that showed contrasting expression trends were involved in different pathways, such as plant hormone signal transduction, glycerolipid metabolism, and phenylpropanoid biosynthesis. The genes enriched in expression profiles 23 and 10 were involved in starch and sucrose metabolism, amino sugar and nucleotide sugar metabolism, and galactose metabolism, and thus may also contribute to improvement of fiber quality. Co-expression network analysis identified five modules that were highly associated with developmental stages of elite-quality fiber. These findings have revealed promising candidate genes for improvement of fiber quality in breeding of upland cotton.

## Figures and Tables

**Figure 1 genes-10-00119-f001:**
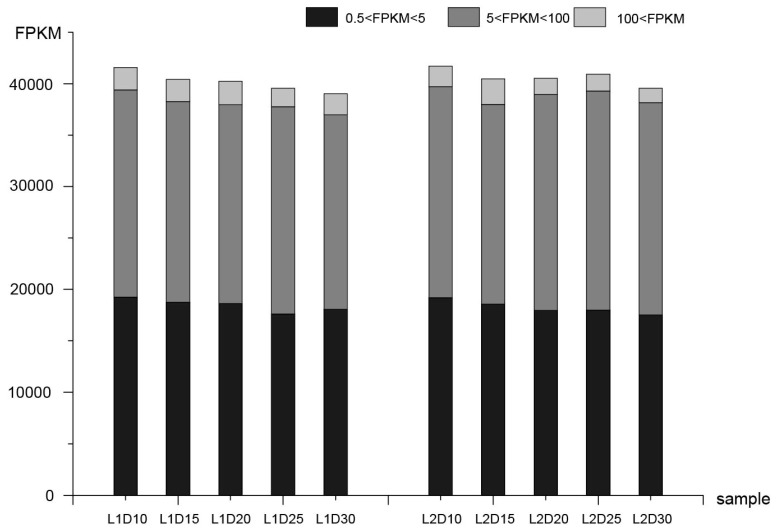
Statistics for transcript levels at each development stage, the numbers of genes and classification were divided by 0.5 < FPKM < 5, 5 < FPKM < 100, and 100 < FPKM in each sample. FPKM: fragments per kilobase of exon per million reads.

**Figure 2 genes-10-00119-f002:**
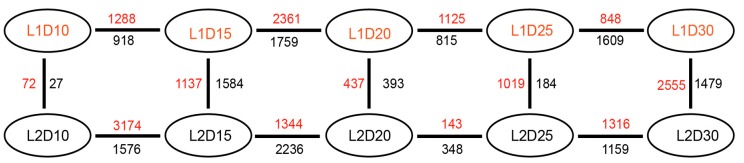
Vertical of L1 relative to L2 and horizontal comparisons of genes that showed differential expression levels between the two lines at the same developmental stage and between different stages of an individual line. Red numbers represent upregulated genes; black numbers indicate down-regulated genes. The differentially expressed genes were identified by the criteria FPKM > 0.5, |log_2_(FC)| > 1.

**Figure 3 genes-10-00119-f003:**
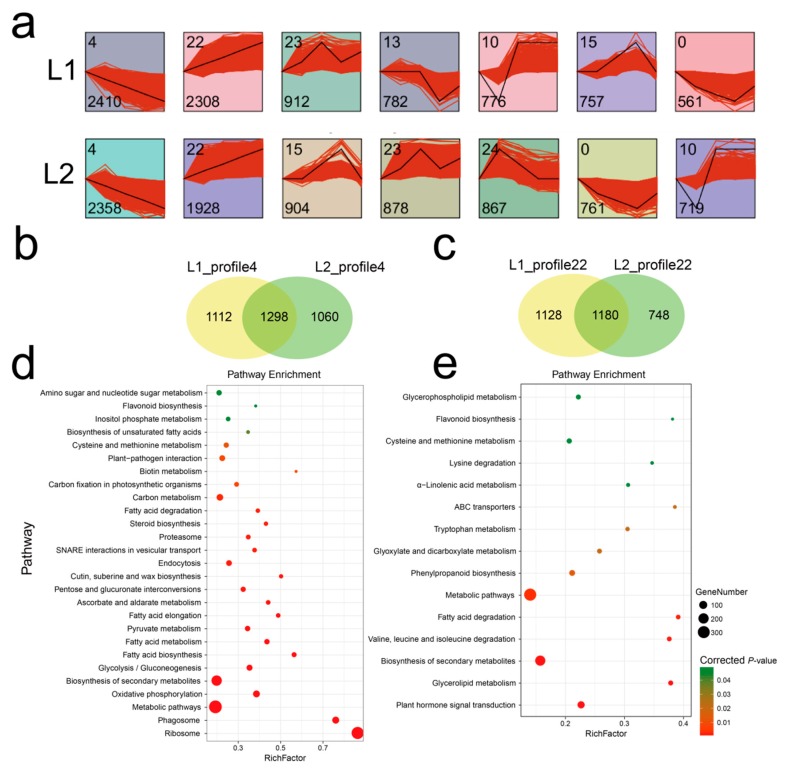
(**a**) Different gene expression profiles in the two recombinant inbred lines (RILs). Each square represents a profile of gene expression trend. The profile ID and gene number in the corresponding profile are shown in the top and bottom left corners, respectively. (**b**) and (**c**) Venn diagram showed the different genes set between L1 and L2 in profile 4 and profile 22, respectively. (**d**) and (**e**) Kyoto Encyclopedia of Genes and Genomes (KEGG) pathways of common genes between L1 and L2 in expression profile 4 and 22. The size of the ball represents the genes number enriched in the pathway; the depth of the color represents the size of the *p*-value.

**Figure 4 genes-10-00119-f004:**
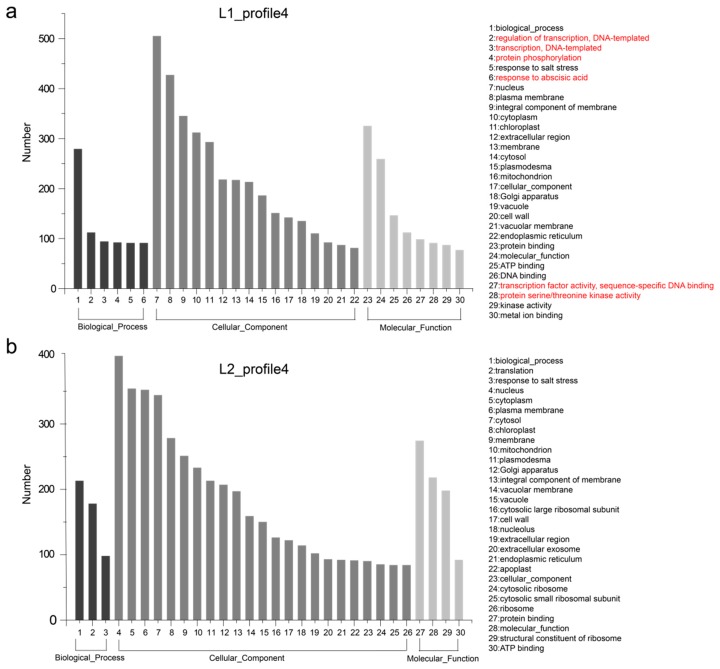
Gene ontology (GO) enrichment of genes listed in expression profile 4. (**a**) and (**b**) show the top 30 terms of GO enrichment for 1112 and 1060 genes unique to L1 and L2, respectively. The main different terms are highlighted in red.

**Figure 5 genes-10-00119-f005:**
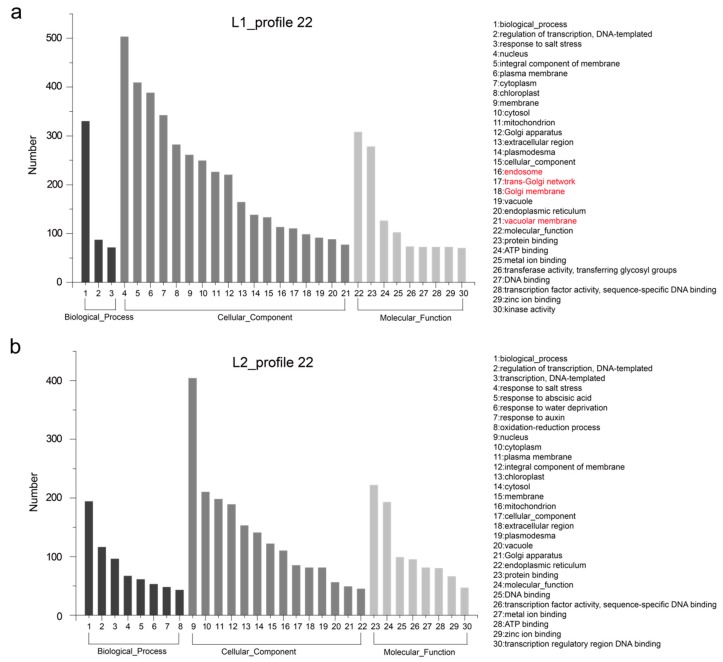
GO enrichment of genes listed in expression profile 22. (**a**) and (**b**) show the top 30 terms of GO enrichment for 1128 and 748 genes unique to L1 and L2, respectively. The main different terms names are highlighted in red.

**Figure 6 genes-10-00119-f006:**
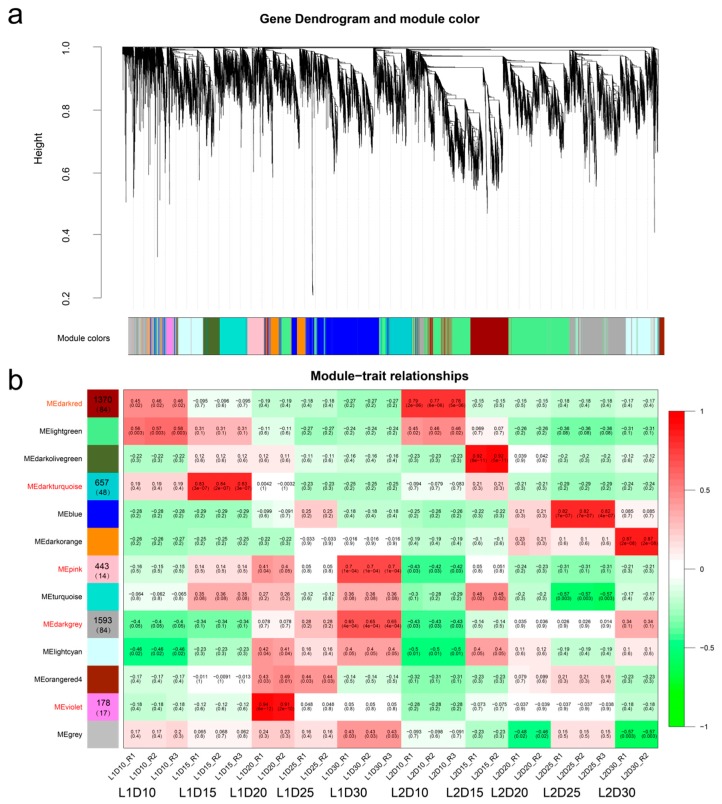
Weighted gene co-expression network analysis (WGCNA) of differentially expressed genes (DEGs) in L1 and L2 at five time points of fiber development. (**a**) Hierarchical dendrogram showing co-expression modules identified by WGCNA. Each leaf in the tree represents one gene. The major tree was divided into 13 modules based on calculation of eigengenes; each module is highlighted in a different color. (**b**) Module–sample relationships. Each row represents a module, and the correlation coefficient and the e-value are shown in each square. Each column corresponds to a sample tissue and replication (R). The number of genes and transcription factors enriched in the module are shown in the left module squares and parentheses, respectively. The modules names in red were highly associated with the high-quality fiber line (L1).

**Figure 7 genes-10-00119-f007:**
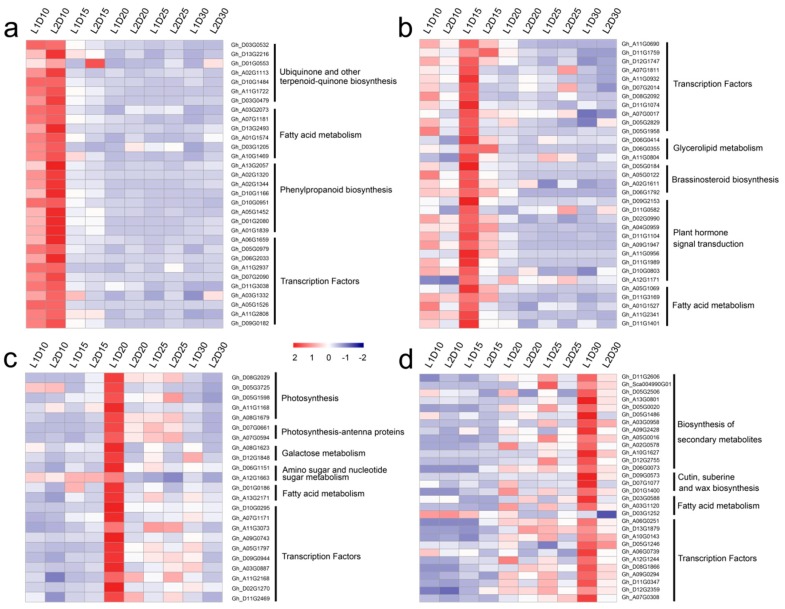
Heatmap comparison DEGs associated with fiber developmental stages. (**a**), (**b**), (**c**), and (**d**) show the DEGs associated to the overrepresented functional categories and transcription factors at 10, 15, 20, and 30 days post-anthesis (DPA), respectively.

**Figure 8 genes-10-00119-f008:**
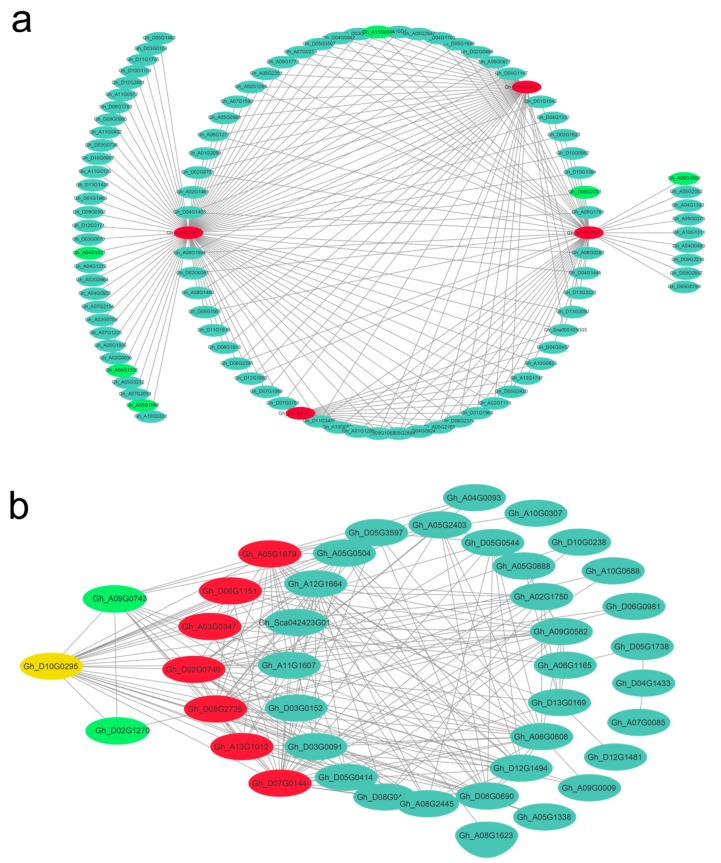
Correlation networks for the dark red and violet modules. Hub genes are indicated by red circles; green circles indicate transcription factors; yellow circles indicate both a hub gene and a transcription factor.

**Table 1 genes-10-00119-t001:** Fragments per kilobase of exon per million reads (FPKMs) and functional categories of genes significantly up-regulated in L1 on 10DPA.

Gene Name	L1D10	L2D10	Description in *Arabidopsis thaliana*
**Transcription factors**			
Gh_A11G0692	6.49	0.9	homeobox-leucine zipper protein 4
Gh_D07G0362	8.39	1.6	Integrase-type DNA-binding superfamily protein
Gh_D10G0270	49.31	11.65	homeobox-3
**Arginine biosynthesis**			
Gh_A05G2143	11.59	2	arginase
Gh_A11G2354	6.86	1.47	glutamate dehydrogenase 1
**Brassinosteroid biosynthesis**			
Gh_D05G0184	19.25	2.36	Cytochrome P450 superfamily protein
Gh_A05G0122	41.06	11.02	Cytochrome P450 superfamily protein
**Metabolic pathways**			
Gh_A03G0507	19.78	1.02	Glycosyl hydrolases family 32 protein
Gh_A05G3473	66.67	5.21	10-formyltetrahydrofolate synthetase
Gh_A05G3997	54.47	5.12	4-coumarate:CoA ligase 2
Gh_A03G1938	10.11	1.62	N-acetylglucosamine-1-phosphate uridylyltransferase 2
Gh_A05G0479	5.74	1.09	aldehyde dehydrogenase 11A3
Gh_D07G0692	13.39	2.57	UDP-glucose 6-dehydrogenase family protein
Gh_D02G0761	9.2	1.95	Thiamin diphosphate-binding fold (THDP-binding) protein
Gh_A10G2327	7.37	1.75	ACT domain-containing small subunit of acetolactate synthase protein
Gh_A09G2449	20.78	5.87	NAD(P)-binding Rossmann-fold superfamily protein
**mRNA surveillance pathway**			
Gh_A05G0238	6.55	0.21	homolog of CFIM-25
Gh_A03G1654	133.17	39.45	poly(A) binding protein 2
Gh_D02G2070	108.02	33.65	poly(A) binding protein 2
**Plant hormone signal transduction**			
Gh_D11G1989	15.5	5.38	Auxin-responsive GH3 family protein
Gh_D09G1585	14.32	1.17	PYR1-like 4
Gh_A09G2421	11.06	1.06	PYR1-like 4

**Table 2 genes-10-00119-t002:** FPKMs and functional categories of genes significantly down-regulated in L1 on 10DPA.

Gene Name	L1D10	L2D10	Description in *A. thaliana*
**Phenylalanine, tyrosine and tryptophan biosynthesis**			
Gh_A13G0258	1.59	12.58	aspartate aminotransferase
Gh_A13G0603	1.36	7.68	3-deoxy-d-arabino-heptulosonate 7-phosphate synthase
**Biosynthesis of amino acids**			
Gh_A07G0619	0.42	9.8	3-phosphoserine phosphatase
**Nicotinate and nicotinamide metabolism**			
Gh_D01G1478	1.61	6.54	quinolinate phoshoribosyltransferase
Gh_A06G0128	5.31	48.44	GA requiring 3
**Phagosome**			
Gh_D13G1883	11.97	96.74	calreticulin 1a
**Transcription factors**			
Gh_A05G0528	2.73	11.38	WRKY DNA-binding protein 11
Gh_A13G1728	9.91	46.36	Zinc finger C-x8-C-x5-C-x3-H type family protein

**Table 3 genes-10-00119-t003:** FPKMs and functional categories of genes significantly upregulated in L1 on 20DPA.

Genes id	L1D20	L2D20	Description
**Biosynthesis of unsaturated fatty acids**			
Gh_Sca007938G01	10.65	1.5	beta-ketoacyl reductase 1
Gh_D01G0186	139.55	29.59	acyl-CoA oxidase 3
Gh_A13G2171	112.41	36.72	acyl-CoA oxidase 4
**Butanoate metabolism**			
Gh_D12G0436	13.79	1.9	short-chain dehydrogenase-reductase B
Gh_A12G1414	56.97	16.99	glutamate decarboxylase 4
Gh_A07G0810	110.4	46.02	glyoxylate reductase 1
**Cutin, suberine and wax biosynthesis**			
Gh_A07G0991	5.36	0.72	Jojoba acyl CoA reductase-related male sterility protein
Gh_D05G1294	67.39	11.9	Glucose-methanol-choline (GMC) oxidoreductase family protein
Gh_D01G1400	172.1	49	Caleosin-related family protein
**Fatty acid elongation**			
Gh_A01G1563	150.26	29.07	3-ketoacyl-CoA synthase 6
Gh_D01G1810	213.82	54.5	3-ketoacyl-CoA synthase 6
Gh_A09G0749	19.87	7.06	3-ketoacyl-acyl carrier protein synthase I
Gh_Sca006141G01	23.73	8.81	acetyl-CoA carboxylase carboxyl transferase subunit β
**Phagosome**			
Gh_A08G2381	635.59	71.34	β-6 tubulin
Gh_A11G2095	1198.32	173.46	tubulin α-3
Gh_Sca012883G01	2274.54	359.05	tubulin β 8
Gh_D08G1960	460.89	75.32	β-6 tubulin
Gh_D05G1052	572.74	176.55	tubulin β 8
Gh_A02G0819	357.99	136.45	tubulin α-2 chain
Gh_D13G1047	259.25	106.17	tubulin α-3
**Transcription factors**			
Gh_D12G0607	16.71	4.38	basic helix-loop-helix (bHLH) DNA-binding superfamily protein
Gh_A07G1171	82.82	17.27	basic region/leucine zipper motif 53
Gh_D05G0463	37.9	13.35	CCCH-type zinc finger family protein
Gh_A12G0561	40.16	15.17	B-box zinc finger family protein
Gh_D05G2596	267.05	106.76	K-box region and MADS-box transcription factor family protein
Gh_A12G1244	21.19	4.2	MYB-like 102
Gh_D08G0157	10.98	2.96	AP2/B3 transcription factor family protein
Gh_D02G0043	40.69	11.26	WRKY family transcription factor family protein

**Table 4 genes-10-00119-t004:** FPKMs and functional categories of genes significantly downregulated in L1 on 20DPA.

Genes Name	L1D20	L2D20	Description in *A. thaliana*
**Biosynthesis of secondary metabolites**			
Gh_D06G0096	0.12	12.6	cytochrome P450, family 82, subfamily C, polypeptide 4
Gh_D04G0605	0.38	9.89	glycerol-3-phosphate acyltransferase 5
Gh_A13G0603	1.47	10.65	3-deoxy-d-arabino-heptulosonate 7-phosphate synthase
Gh_D05G2554	4.26	22.71	acetyl-CoA carboxylase 1
**Nitrogen metabolism**			
Gh_Sca069862G01	18.38	189.16	glutamine synthase clone R1
Gh_D07G1773	54.01	333.88	glutamine synthase clone R1
Gh_A01G1586	1.4	7.56	nitrate reductase 2
Gh_D01G1872	7.83	26.87	nitrate reductase 2
**Phenylpropanoid biosynthesis**			
Gh_D10G0473	11.3	63.26	4-coumarate:CoA ligase 1
Gh_D08G1135	15.62	62.86	O-methyltransferase 1
Gh_A08G0711	32.62	105.85	Peroxidase superfamily protein
Gh_A04G1032	29.29	80.06	S-adenosyl-L-methionine-dependent methyltransferases
**Zeatin biosynthesis**			
Gh_D05G1813	9.23	43.01	cytokinin oxidase 5
Gh_A05G0290	6.64	25.84	cytokinin oxidase 7
Gh_A05G1631	7.6	26.27	cytokinin oxidase 5
Gh_D05G0391	12.27	39.85	cytokinin oxidase 7
**Transcription factors**			
Gh_A02G1575	9.12	28.68	GRAS family transcription factor
Gh_D01G0974	102.38	371.7	basic leucine-zipper 44
Gh_A11G2875	3.42	15.35	myb-like transcription factor family protein
Gh_A11G2522	3.52	19.11	myb domain protein 120
Gh_A05G3778	2.5	24.64	LOB domain-containing protein 13
Gh_D07G1330	0.53	5.69	NAC domain transcriptional regulator protein
Gh_D06G0254	0.52	6.37	GATA-type zinc finger protein with TIFY domain
Gh_A10G0516	2.06	26.94	myb domain protein 26
Gh_D08G1424	0.11	6.98	WRKY DNA-binding protein 3

**Table 5 genes-10-00119-t005:** Candidate hub genes in 10, 15, 20, and 30 DPA modules.

Gene Name	K_ME_	*Arabidopsis* ID	Function Description in *Arabidopsis thaliana* Gene
10 DPA specific darkred module			
Gh_D12G2172	0.991	AT1G12240	Glycosyl hydrolases family 32 protein
Gh_A10G1961	0.991	AT2G37620	actin 1
Gh_A11G1087	0.997	AT4G03100	Rho GTPase activating protein with PAK-box [46]
Gh_D06G0618	0.991	AT4G28950	RHO-related protein from plants 9
Gh_D05G0591	0.992	AT4G31890	ARM repeat superfamily protein
15 DPA of L1 specific darkturquoise module			
Gh_A02G1692	0.968	AT3G51430	Calcium-dependent phosphotriesterase superfamily protein
Gh_A07G0225	0.983	AT4G32330	TPX2 (targeting protein for Xklp2) protein family [47,48]
Gh_A12G0226	0.971	AT2G15780	Cupredoxin superfamily protein
Gh_D03G1054	0.973	AT1G01630	Sec14p-like phosphatidylinositol transfer family protein
Gh_D12G0026	0.964	AT3G51895	sulfate transporter 3;1
20 DPA of L1 specific violet module			
Gh_A03G0347	0.967	AT2G28790	Pathogenesis-related thaumatin superfamily protein
Gh_A05G1879	0.981	AT4G33580	beta carbonic anhydrase 5
Gh_A13G1012	0.965	AT4G09510	cytosolic invertase 2
Gh_D06G1151	0.968	AT4G16130	arabinose kinase
Gh_D10G0295	0.974	AT1G68810	basic helix-loop-helix (bHLH) DNA-binding superfamily protein 30 [49]
Gh_D02G0740	0.97	AT1G21460	Nodulin MtN3 family protein
Gh_D07G0144	0.969	AT5G20860	Plant invertase/pectin methylesterase inhibitor superfamily [50]
Gh_D08G2725	0.973	AT3G62690	AtL5
20&30 DPA of L1 specific pink module			
Gh_A08G1486	0.95	AT5G47120	BAX inhibitor 1
Gh_A08G2508	0.944	AT4G27000	RNA-binding (RRM/RBD/RNP motifs) family protein
Gh_A11G2726	0.961	AT3G13540	myb domain protein 5 [51,52]
Gh_D03G1843	0.958	AT1G32050	SCAMP family protein
Gh_D09G0347	0.962	AT5G14240	Thioredoxin superfamily protein
30 DPA of L1 specific darkgrey module			
Gh_A09G1137	0.981	AT5G04160	UDP-URONIC acid transporter 1 [53,54];
Gh_A10G1754	0.989	AT5G06390	FASCICLIN-like arabinogalactan protein 17 precursor
Gh_D01G1697	0.982	AT1G20550	O-fucosyltransferase family protein
Gh_D08G2152	0.985	AT2G33170	Leucine-rich repeat receptor-like protein kinase family protein [55]
Gh_D10G1056	0.99	AT5G64030	S-adenosyl-L-methionine-dependent methyltransferases superfamily protein [56]

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
