# Peer review of "Co-Expression Network Analysis and Hub Gene Selection for High-Quality Fiber in Upland Cotton (Gossypium hirsutum) Using RNA Sequencing Analysis"

_genes, 2019, doi:10.3390/genes10020119_

Round 1
Reviewer 1 Report
Good work. I have read a similar work before but I think that is on a different plant. Well writing paper. I hope the diagram will be more visible in the final print out. Otherwise what will be the point of a diagram that is not readable?
Thank you.
Author Response
Response to Reviewer 2 Comments
Point 1: My major concern is negligence of down-regulated genes in the present analysis. Authors have ignored the fact that a down-regulation od set of genes are also part of the regulatory mechanism and can be important for fiber development. Adding the analysis of those genes would definitely add value to the data. 
Response 1: We are very grateful to you for your valuable comments on our manuscript review work. In the section “3.2 Differential gene expression analysis”, we add two tables that contained the representative down-regulated significantly genes in L1 relative to L2 on 10 DPA and 20 DPA, which named Table 2 and Table 4, simultaneously, we changed the “Table 2” and “ Table 3” in the previous manuscript to “Table 3” and “ Table 5”. These table number changes were shown in line 197, 199, 214, 216, and 354. The results for down-regulated genes were shown in line 199-200 and 216-217. 
Point 2: Line 66: Is use of “strain” appropriate for plants?
Response 2: We investigated plant research literatures, which showed that the word of “strain” could be used in plant. And the biology meaning of strain is variants of plants, viruses or bacteria; or an inbred animal used for experimental purposes, in WIKIPEDIA (https://en.wikipedia.org/wiki/Strain). Therefore, we think that the use of “strain” is appropriate for plant.
Point 3: Bowtie 2.0.6 is mentioned in the methods section (line 96) whereas result section 3.1 it’s Bowtie 1 (line 131)
Response 3: The purposes of Bowtie 2.0.6 and Bowtie 1 are different in the method. Bowtie 1.0 is used to obtain the reads positon, then we can show the density of reads on chromosomes. However, the Bowtie 2.0.6 is used to construct the index files with the reference genome sequence.
Point 4: Line 106 says about analysis criteria of FPKM>5 whereas line 155 says FPKM>0.5. Please correct
Response 4: The phrase “FPKM>5” is a mistake in manuscript writing. We thank you for your commend and have revised it in line 108.
Point 5: Except for figure 3, all figures should be increased in size in order to understand the data.
Response 5: We have increased the size of all the figures by 1.5 times.
Point 6: Define axis details for figure 2A
Response 6: We have added the definition of the axis detail.
Thank you again!!

Reviewer 2 Report
Zou et al have performed a good comparison of RNA-seq data from two contrasting fiber quality cotton RILs and have identified HUB genes. The manuscript is well written and following a few suggestions might help in further improvement:
1. My major concern is negligence of down-regulated genes in the present analysis. Authors have ignored the fact that a down-regulation od set of genes are also part of the regulatory mechanism and can be important for fiber development. Adding the analysis of those genes would definitely add value to the data.
2. Line 66: Is use of “strain” appropriate for plants?
3.      Bowtie 2.0.6 is mentioned in the methods section (line 96) whereas result section 3.1 it’s Bowtie 1 (line 131).
4. Line 106 says about analysis criteria of FPKM>5 whereas line 155 says FPKM>0.5. Please correct.
5. Except for figure 3, all figures should be increased in size in order to understand the data.
6. Define axis details for figure 2A.
Author Response

(The authors gave the same response as above.)

Reviewer 3 Report
The authors have performed an adequate experiment. They have performed several analytical approaches and provided with a huge amount of interesting data. However, I consider that the writing of this report shows many flaws, specially, regarding the further interpretation of the results, which is practically missing. I encourage the authors to make a further effort in inferring the outcome of this research study and reorganize the manuscript. However, here you can find my comments about the weaknesses, flaws or minor errors I could detect in the report. I hope my suggestions to be helpful.
2.1 Plant materials:
-Data accession is highlited in yellow
3.1 Transcriptome sequencing analysis:
-Figure 1 is illustrative but not necessary. Doesn't provides with relevant information not contained in the text. In addition, in line 132 you say that Fig1 c,d indicate the difference of level of transcripts among the fiber development stages. Well, I don't see the differences (maybe because the size of the figures). Anyway, the real differences are presented in paragraph 3.3. I recommend to consider removing fig a,b,e in order to make c and d bigger or removing fig1 in order to shorten the paper.
-Fig2 a has two L1 lines. One has to be L2 (misspelling error).Further, to my opinion, figure 2 a is not needed either. Regarding Fig2b,I would need an explanation in the text about why that information is relevant for the research. You don't focus on shared/not shared genes among developmental stages but compared the differential expression in 3.2 and 3.4. Therefore I assume that Fig2b is merely illustrative and in that case 5 venn diagrams comparing the L1 and L2 or a table with all the comparisons would provide with more interesting information related to the scope of this study.
3.2 Differential gene expression analysis:
-I suggest to clarify in both the text (line 147) and figure 3 legend that the comparison is "L1 relative to L2 (poor quality fiber line). In this way, It will be clarified at the beginning and in the figure and it won't be necessary to repeat it in the following four paragraphs.
-line 165: misspelled "iin L2"
-I would appreciate an explanation about how the "Representative significantly DEGs" for tables 1 and 2 were chosen. Was it because highest log2(FC)? Was it because those genes were exclusively up-regulated in that particular development stage and not at the other four? Was it because those genes are not included in the 363 DEGS that are also included in the 16 stable QTLs?
-On the other hand, it is not clear why you made tables for 10dpa and 20 dpa but not for 15, 25 and 30 dpa? Did you focused on those time points because represent the fiber elongation and the onset of the secondary wall biosynthesis which determine fiber strength and fineness? If so, that should be explained in the text. In any case, once you highlight the relevance of certain genes up-regulated (none is downregulated in the table), the FPKM at the other time points could also fit into the table and explain why those genes are relevant in the discussion.
-line 174. Check English. "genes were showed/ genes showed..."
-lines 188-194. It is stated that horizontal comparison is especially different for 15vs20 and 20vs25 because the number of up-regulated genes in L1 are 1.75 and 5.5-fold that in L2 respectively. However, it is also 2.46 and 1.55 fold at 10vs15 and 25vs30 but in this case, is L2 who has more up-regulated genes. You should rewrite the paragraph to make your point about the horizontal comparisons.
3.3. Congruence analysis
-To my opinion this is an interesting analysis but no further analysis have been made on the 363 DEGs located in previously identified stable QTLs. They are simply reported in table S6 but there is a lack of interpretation of the results in the discussion. There is not even a connection with the rest of the results reported.
-Table S6: In the header is called table S8 and space is needed in "homologousgenes". Again, I recommend to include the FPKM at the other developmental stages too.
3.4 Temporal gene expression patterns
-In lines214-217 there is an incongruence since it states that "half of the genes were different in each expression pattern even the number of genes were similar" but in line 214 it is said that L1 has 19.7% more of genes that L2 in expression profile 22. It should be clarified that is in the expression profile 4 where the numbers are similar. However, what is important is that the genes that are different between lines. In addition, further information about those unique genes in each line like whether the are expressed or not in the other line and if they are, whether the expression profile is very different in the other line.
-From the context I inferred that you focused on profiles 4 and 22 because they contained the largest number of genes covering more that 50% of the genes (around 8500) and also maybe because they represent the most clear (continuous trend) and contrasting profiles (up and down). I also found out that they are the only 2 profiles with e-value=0.0 (table S7). In any case, the reason for focusing on these profiles for functional analysis (GO enrichment) should be clarified in the text.
-Figure 4 legend should clarify in Fig4a which row of profiles corresponds to L1 and L2. In addition, it lacks information about 4b, 4c and 4d.
-Line 246-247: It is a question of semantics; STEMs analyses the enrichment of the clusters in certain GO terms but I believe that the genes are not enriched. Please be careful with the construction of this tense and also in the abstract (line 25).
- lines 248-261. the profiles 32 and 10 are also functionally analyzed but there it is not explain if there are differences in those profiles between L1 and L2 in genes (shared/not shared) or GO terms as it is done for profiles 4 and 22.
- All in all, there is a lack of discussion of the results obtained in paragraph 3.4 as well as for 3.3. These results are not even mentioned in the discussion.
3.5 Gene co-expression network analysis
-Line 270. "single fiber tissue sample of L1" could be misinterpreted as "one replicate" instead of "a developmental stage only in L1"
-Figure5b and figure 6 are very hard to see printed and loose resolution when zooming the pdf but if the quality is good on the html version this is not a big issue. In any case, in fig6a the first pathway is misspelled "erpenoid-quinone synthesis" In figure 5b the name of the first of the five modules highly associated with the high-quality fiber (dark red module) looks like orange instead of red. In addition, since I can not read the correlation coefficients and e-values I would request to clarify the threshold for filtering the highly associated modules on the basis of their coefficient between modules and tissues in materials and methods.
-Table S9 header is "table S6".
-Figure6 legend: Another question of semantics; "...show the functional categories of the over-represented DEGs and transcriptions factors.." I suggest something like "...show the DEGs associated to the over-represented functional categories and transcription factors..."
-Table 3. It should include the threshold Kme for the genes of the five modules to be considered top-ranked and therefore hub genes. This information can be either here or in materials and methods.
-Table S10 header is "tableS7". Please check re-numbering of the supplementary data.
4. Discussion
- 338-339. I consider that "incorporating additional developmental stages compared to previous RNA-Seq studies" it is a vague discussion of your results compared to Li et al., 2017 and Islam et al., 2016 and Zhang et al., 2017 results. The authors should be more precise in the knowledge provided by their work compared to these recent previous works.
-All in all the section 4.1 and the first paragraph at 4.2 just repeat the results described in 3.1 and 3.2. These paragraphs could be shortened and fusion so there is more space for discussion of the results. Further, any discussion about the results described in 3.3 and 3.4 is missing.
- Lines368-377 looks like a reading of table 3 and the papers where these genes have been found are mentioned between brackets. These references could be included in table 3 and leave more space for a proper discussion about the significance of the results found.
5. Conclusion
The conclusion section is not mandatory. However, the reason to be included is to collate the conclusions after a long discussion. Therefore, this section should summarize the findings obtained in terms of knowledge acquired. On the contrary, lines 379-387 summarize again the analysis performed and lines 387-394 represents a part that was missing in the discussion. All in all, I recommend a further interpretation of the results obtained in an attempt of relating the results obtained by the different analysis (vertical and horizontal comparisons, genes in QTLs, temporal expression patterns and hub genes), re-write the discussion and summarizing the concluding remarks in the conclusion section.

Author Response
Response to Reviewer 3 Comments
Firstly, as one of the co-authors, I sincerely thank you for your review and valuable comments. Thank you very much!
Point 1: 2.1 Plant materials:  --Data accession is highlited in yellow 
Response 1: We have changed the color of this sentence.
Point 2: 3.1 Transcriptome sequencing analysis:  --Figure 1 is illustrative but not necessary. Doesn't provides with relevant information not contained in the text. In addition, in line 132 you say that Fig1 c,d indicate the difference of level of transcripts among the fiber development stages. Well, I don't see the differences (maybe because the size of the figures). Anyway, the real differences are presented in paragraph 3.3. I recommend to consider removing fig a,b,e in order to make c and d bigger or removing fig1 in order to shorten the paper.
Response 2: We think the Figure 1 is kind of visualization of data and still need for the manuscript. According to your suggestion, we divided the Figure1 into two pictures as Figure S1 and S2 to enhance the resolution and shorten the paper. 
Point 3: -Fig2 a has two L1 lines. One has to be L2 (misspelling error).Further, to my opinion, figure 2 a is not needed either. Regarding Fig2b,I would need an explanation in the text about why that information is relevant for the research. You don't focus on shared/not shared genes among developmental stages but compared the differential expression in 3.2 and 3.4. Therefore I assume that Fig2b is merely illustrative and in that case 5 venn diagrams comparing the L1 and L2 or a table with all the comparisons would provide with more interesting information related to the scope of this study. 
Response 3: Thank you for your reminder of the misspelling. We have revised it. For the figure 2b, we deleted it. 
Point 4: 3.2 Differential gene expression analysis:   --I suggest to clarify in both the text (line 147) and figure 3 legend that the comparison is "L1 relative to L2 (poor quality fiber line). In this way, It will be clarified at the beginning and in the figure and it won't be necessary to repeat it in the following four paragraphs.  
Response 4: we have revised the legend in line 156 and delete this phrase in the following four paragraphs in line 162, 180, 202, and 205. 
Point 5:-line 165: misspelled "iin L2".
Response 5: The "iin L2" has been changed to "in L2" in line 172. Thank you sincerely.
Point 6: -I would appreciate an explanation about how the "Representative significantly DEGs" for tables 1 and 2 were chosen. Was it because highest log2(FC)? Was it because those genes were exclusively up-regulated in that particular development stage and not at the other four? Was it because those genes are not included in the 363 DEGS that are also included in the 16 stable QTLs?
Response 6: We have added the reason to explain the representative genes in lines 162-164. Representative significantly DEGs are not only showed high |log2(FC)| between L1 and L2, but also significantly (corrected p-value < 0.01)) enriched in the KEGG pathways. In fact, due to the size limitation of the table, some genes that are also significantly enriched in KEGG pathways and with high |log2(FC)| values are not listed in Tables 1-4, but they are also reflected in the supplementary tables.
Point 7: -On the other hand, it is not clear why you made tables for 10dpa and 20 dpa but not for 15, 25 and 30 dpa? Did you focused on those time points because represent the fiber elongation and the onset of the secondary wall biosynthesis which determine fiber strength and fineness? If so, that should be explained in the text. In any case, once you highlight the relevance of certain genes up-regulated (none is downregulated in the table), the FPKM at the other time points could also fit into the table and explain why those genes are relevant in the discussion.
Response 7: We listed the 10 and 20 DPA because Wendel et al. have selected this two DPA to explore the expression profiles between cultivar and wild upland cotton as described in line 377-378. In order to facilitate the visual comparison of the difference in expression between two lines in the same stage, we still think that it is better to provide only FPKM values of one stage of the two lines in Table1-4. 
Point 8: -line 174. Check English. "genes were showed/ genes showed..."  
Response 8: we have revised it in line 185. Thank you!
Point 9: -lines 188-194. It is stated that horizontal comparison is especially different for 15vs20 and 20vs25 because the number of up-regulated genes in L1 are 1.75 and 5.5-fold that in L2 respectively. However, it is also 2.46 and 1.55 fold at 10vs15 and 25vs30 but in this case, is L2 who has more up-regulated genes. You should rewrite the paragraph to make your point about the horizontal comparisons.
Response 9: We have realized that this quantitative comparison is vague. So we deleted it because we think the STEM analysis in section 3.4 was kind of horizontal comparison.
Point 10: 3.3. Congruence analysis    --To my opinion this is an interesting analysis but no further analysis have been made on the 363 DEGs located in previously identified stable QTLs. They are simply reported in table S6 but there is a lack of interpretation of the results in the discussion. There is not even a connection with the rest of the results reported.
Response 10: We have revised this paragraph in line 225-234, in which we emphasize the importance of these genes. What’s more, several genes are also located in previous studies, which further demonstrate that these genes play important roles in fiber development.
Point 11: Table S6: In the header is called table S8 and space is needed in "homologousgenes". Again, I recommend to include the FPKM at the other developmental stages too.
Response 11: We have revised the number and the spelling of homolo gousgenes. The FPKM at other time points were also added in Table S6.
Point 12: 3.4 Temporal gene expression patterns  --In lines214-217 there is an incongruence since it states that "half of the genes were different in each expression pattern even the number of genes were similar" but in line 214 it is said that L1 has 19.7% more of genes that L2 in expression profile 22. It should be clarified that is in the expression profile 4 where the numbers are similar. However, what is important is that the genes that are different between lines. In addition, further information about those unique genes in each line like whether the are expressed or not in the other line and if they are, whether the expression profile is very different in the other line.  
Response 12: We have clarified that is in the expression profile 4 where the numbers are similar in line 245, according to your suggestion, thank you very much! The further information about genes unique to L1 and L2 in these profiles have performed using GO annotation as shown in figure 4 and 5, figure S3 and S4. The functional differences between genes unique to L1 and L2 in profile 4 and 10 were described in lines 262-275. The functional differences between genes unique to L1 and L2 in profile 10 and 23 were described in lines 304-316. 
Point 13:-From the context I inferred that you focused on profiles 4 and 22 because they contained the largest number of genes covering more that 50% of the genes (around 8500) and also maybe because they represent the most clear (continuous trend) and contrasting profiles (up and down). I also found out that they are the only 2 profiles with e-value=0.0 (table S7). In any case, the reason for focusing on these profiles for functional analysis (GO enrichment) should be clarified in the text.
Response 13: The figure s1 and s2 in original manuscript were used to show GO analysis. According to your suggestion, we changed these two pictures to figure 4 and 5 in the text. 
Point 14: -Figure 4 legend should clarify in Fig4a which row of profiles corresponds to L1 and L2. In addition, it lacks information about 4b, 4c and 4d.
Response 14: Thank you for your reminder, we have added the detail legend of figure 3 in lines 149-255. 
Point 15: -Line 246-247: It is a question of semantics; STEMs analyses the enrichment of the clusters in certain GO terms but I believe that the genes are not enriched. Please be careful with the construction of this tense and also in the abstract (line 25).
Response 15: Thank you for your reminder, we have revised this semantics in lines 288-289 and line 25.
Point 16: - lines 248-261. the profiles 32 and 10 are also functionally analyzed but there it is not explain if there are differences in those profiles between L1 and L2 in genes (shared/not shared) or GO terms as it is done for profiles 4 and 22.
Response 16: we have performed the GO analysis of genes unique to L1 and L2 in this two profiles as shown in figure s3 and s4, and described in lines 304-316.
Point 17: - All in all, there is a lack of discussion of the results obtained in paragraph 3.4 as well as for 3.3. These results are not even mentioned in the discussion.
Response 17: We have discussed the results of section 3.3 and 3.4 in lines 439-448. 
Point 18: 3.5 Gene co-expression network analysis   --Line 270. "single fiber tissue sample of L1" could be misinterpreted as "one replicate" instead of "a developmental stage only in L1"
Response 18: We have revised it as “certain developmental stage”, thank you!
Point 19: -Figure5b and figure 6 are very hard to see printed and loose resolution when zooming the pdf but if the quality is good on the html version this is not a big issue. In any case, in fig6a the first pathway is misspelled "erpenoid-quinone synthesis" In figure 5b the name of the first of the five modules highly associated with the high-quality fiber (dark red module) looks like orange instead of red. In addition, since I can not read the correlation coefficients and e-values I would request to clarify the threshold for filtering the highly associated modules on the basis of their coefficient between modules and tissues in materials and methods.
Response 19: (1) We have increased the size of picture by 1.5 times. (2)The "erpenoid-quinone synthesis" has been corrected as shown in figure 7 line 348. (3) the color names were generated by WGCNA package. (4) We have described the methods of selecting modules for further analysis in lines 120-121..
Point 20: -Table S9 header is "table S6"
Response 20: We have revised this error. Thank you very much!
Point 21: -Figure6 legend: Another question of semantics; "...show the functional categories of the over-represented DEGs and transcriptions factors.." I suggest something like "...show the DEGs associated to the over-represented functional categories and transcription factors..."
Response 21: We have revised this sentence in lines 351-353, according to your suggestion.
Point 22: -Table 3. It should include the threshold Kme for the genes of the five modules to be considered top-ranked and therefore hub genes. This information can be either here or in materials and methods.  
Response 22: We selected the top five KME genes as hub genes just like previous study (Du et al, 2017), and there is no clear threshold to determine the hub genes in the WGCNA package.
Point 23: Table S10 header is "tableS7". Please check re-numbering of the supplementary data.
Response 23: Thank you for your reminder, we have checked all Table order again.
Point 24: 4. Discussion--338-339. I consider that "incorporating additional developmental stages compared to previous RNA-Seq studies" it is a vague discussion of your results compared to Li et al., 2017 and Islam et al., 2016 and Zhang et al., 2017 results. The authors should be more precise in the knowledge provided by their work compared to these recent previous works.
Response 24: According to your suggestion, we have re-write the discussion. In lines 388-390 and 410-419, we have summarized most of the RNA-seq researches in cotton and discussed our results in lines 428-448 and lines 455-472.
Point 25:-All in all the section 4.1 and the first paragraph at 4.2 just repeat the results described in 3.1 and 3.2. These paragraphs could be shortened and fusion so there is more space for discussion of the results. Further, any discussion about the results described in 3.3 and 3.4 is missing.
Response 25: We have discussed the section 3.3 and 3.4 which introduce the results of genes in ATLs and STEM analysis in lines 439-448.
Point 26:- Lines 368-377 looks like a reading of table 3 and the papers where these genes have been found are mentioned between brackets. These references could be included in table 3 and leave more space for a proper discussion about the significance of the results found.
Response 26: According to your suggestion, we have re-write the discussion. In lines 388-390 and 410-419, we have summarized most of the RNA-seq researches in cotton and discussed our results in lines 428-448 and lines 455-472.
Point 27:5. Conclusion---The conclusion section is not mandatory. However, the reason to be included is to collate the conclusions after a long discussion. Therefore, this section should summarize the findings obtained in terms of knowledge acquired. On the contrary, lines 379-387 summarize again the analysis performed and lines 387-394 represents a part that was missing in the discussion. All in all, I recommend a further interpretation of the results obtained in an attempt of relating the results obtained by the different analysis (vertical and horizontal comparisons, genes in QTLs, temporal expression patterns and hub genes), re-write the discussion and summarizing the concluding remarks in the conclusion section.
Response 27: We have re-summarized the conclusion from the results of multiple comparisons, DEGs in QTLs, STEM analysis and hub genes in lines 489-499.

Round 2
Reviewer 2 Report
Authors have successfully answered all my concerns.